# Effect modification of diabetic status on the association between exposure to particulate matter and cardiac arrhythmias in a general population: A systematic review and meta-analysis

Kiattichat Tassanaviroj[1], Pimchanok Plodpai[1], Pakpoom Wongyikul[2]☯*, Krittai Tanasombatkul[2,3], Krekwit Shinlapawittayatorn[4,5,6], Phichayut Phinyo[2,3]☯*

1 Faculty of Medicine, Chiang Mai University, Chiang Mai, Thailand, 2 Center for Clinical Epidemiology and Clinical Statistics, Faculty of Medicine, Chiang Mai University, Chiang Mai, Thailand, 3 Department of Family Medicine, Faculty of Medicine, Chiang Mai University, Chiang Mai, Thailand, 4 Cardiac Electrophysiology Research and Training Center, Faculty of Medicine, Chiang Mai University, Chiang Mai, Thailand, 5 Center of Excellence in Cardiac Electrophysiology Research, Chiang Mai University, Chiang Mai, Thailand, 6 Cardiac Electrophysiology Unit, Department of Physiology, Faculty of Medicine, Chiang Mai University, Chiang Mai, Thailand

☯ These authors contributed equally to this work.
* phichayutphinyo@gmail.com (PP); aumkidify@gmail.com (PW)

## Abstract

Particulate matter (PM) has various health effects, including cardiovascular diseases. Exposure to PM and a diagnosis of diabetes mellitus (DM) have been associated with an increased risk of cardiac arrhythmias. However, no comprehensive synthesis has been conducted to examine the modifying effect of DM on the association between PM and arrhythmia events. Thus, the objectives of this review were to investigate whether the association of PM is linked to cardiac arrhythmias and whether DM status modifies its effect in the general population. The search was conducted on PubMed/MEDLINE and Embase until January 18, 2023. We included cohort and case-crossover studies reporting the effect of PM exposure on cardiac arrhythmias and examining the role of diabetes as an effect modifier. We used the DerSimonian and Laird random-effects model to calculate the pooled estimates. A total of 217 studies were found and subsequently screened. Nine studies met the inclusion criteria, and five of them were included in the meta-analysis. The participants numbered 4,431,452, with 2,556 having DM. Exposure to PM of any size showed a significant effect on arrhythmias in the overall population (OR 1.10, 95% CI 1.04–1.16). However, the effect modification of DM was not significant (OR 1.18 (95% CI 1.01–1.38) for DM; OR 1.08 (95% CI 1.02–1.14) for non-DM; p-value of subgroup difference = 0.304). Exposure to higher PM concentrations significantly increases cardiac arrhythmias requiring hospital or emergency visits. Although the impact on diabetic individuals is not significant, diabetic patients should still be considered at risk. Further studies with larger sample sizes and low bias are needed.

**Data Availability Statement:** All relevant data are within the manuscript and its Supporting Information files.

**Funding:** This study was partially supported by Chiang Mai University and the Faculty of Medicine, Chiang Mai University. No additional external funding was received for this study.

**Competing interests:** The authors have declared that no competing interests exist.

## Introduction

Exposure to particulate matter (PM) is one of the most important global public health concerns. It leads to a variety of negative health effects, including cardiovascular diseases (CVD) [1]. PM can directly translocate into the bloodstream, deposit onto vascular endothelium, triggering local oxidative stress and inflammation, which finally destabilizes the atherosclerotic plaque and initiates thrombus formation [1, 2]. This PM-induced oxidative stress, together with an increase in reactive oxygen species (ROS) level and a dysregulated cardiac autonomic nervous system, also plays an important role in myocardial injury and cardiac arrhythmias [3–5]. Primarily, exposure to PM increases the risk of ventricular arrhythmias, but there are also reports of atrial fibrillation (AF). A recent systematic review concluded that there was sufficient evidence to confirm that short-term PM exposure influences the occurrence of arrhythmias, atrial fibrillation (AF), and cardiac arrest [6]. However, there is still insufficient evidence to draw conclusions about the long-term effects of PM exposure and arrhythmias. While the risk of these events is elevated in the general population, they occur even more commonly in susceptible individuals, such as those with preexisting cardiac conditions (e.g., congestive heart failure), diabetes, and hypertension [7–9].

It is well established that diabetes mellitus (DM) is a major risk factor for CVD and cardiovascular (CV) death [10]. While evidence regarding the roles of DM in developing coronary artery disease and myocardial disease is quite saturated, relatively less is known about its contribution to cardiac arrhythmogenesis [11]. Specifically, DM alters energy metabolism, increases systemic inflammation, and induces oxidative stress, which leads to structural remodeling in both the atria and ventricles, altering the electrical conduction of the heart. Inflammation also causes defects in mitochondrial dynamics, leading to mitochondrial dysfunction and disrupting cardiac electrophysiology, making diabetic patients more susceptible to both atrial and ventricular arrhythmias [10–13]. To date, more studies have been conducted to draw connections between diabetes and AF than other types of arrhythmias [11]. Notably, the occurrence of arrhythmia, specifically AF, has been found to be associated with both type 1 and type 2 DM [14, 15]. As the pathogenic mechanism of arrhythmia from both PM exposure and DM seemed to overlap, and there was previous evidence showing that exposure to PM increased the risk of arrhythmia in both DM-induced rats and diabetic patients [16, 17], it was hypothesized that the effect of PM on cardiac arrhythmia might be more pronounced in individuals with DM than in individuals without DM [10, 12, 16, 17].

There were meta-analyses that investigated the association between PM and cardiac arrhythmias, but almost all of them focused specifically on AF, which is only one type of cardiac arrhythmia. Furthermore, none of them performed a subgroup analysis of diabetic status [18–22]. Only one meta-analysis was conducted to find the association between PM and AF with the subgroup analysis; they found that exposure to PM was associated with the onset of AF episodes, but the modifying effect of diabetic status was not observed [21]. We then aimed to evaluate the effect of PM on any cardiac arrhythmias and to compare the occurrence of any cardiac arrhythmias in the presence of PM between diabetic and non-diabetic patients.

## Methods

### Study design and conduct

This systematic review and meta-analysis were conducted and reported in accordance with the Cochrane Handbook of Systematic Reviews and the Preferred Reporting Items for Systematic Reviews and Meta-analyses (PRISMA) [23], respectively. The study was registered in PROSPERO [CRD42023407589].

## Eligibility criteria

Studies were included in our systematic review based on specific criteria: (1) a prospective or retrospective longitudinal study design (e.g., cohort or case-control); (2) inclusion of populations with and without DM (regardless of DM types) with separate reporting of outcome data for each group; (3) PM as the exposure of interest; and (4) reporting the occurrence of any cardiac arrhythmias (e.g., atrial fibrillation) as one of the outcomes. The studies eligible for meta-analysis were those providing the odds ratio (OR) as the outcome parameter. It is important to note that cross-sectional studies were not considered eligible for inclusion in our analysis.

Studies were excluded from our analysis based on specific criteria, including: (1) studies published in languages other than English; (2) studies lacking an abstract and/or available full text; and (3) duplicated studies.

## Information sources and search strategy

A pre-specified search strategy was employed to search for literature until the end of December 2022. Electronic medical and scientific databases included PubMed/MEDLINE and EMBASE. The key search terms were "Diabetes mellitus," "Particulate matter," and "cardiac arrhythmias." A search for grey literature or unpublished studies was conducted through Clinical Trial Registry and Google Scholar. The authors also reviewed the list of references from previous systematic reviews and/or meta-analyses with related topics. Relevant studies identified from reference reviews but not included in previously mentioned electronic database searches would also be included at this point. An updated search was conducted on January 18, 2023, before the final analyses and formal dissemination, using the same set of search terms as ascertained in the pre-specified search strategies.

## Study selection

We searched the titles and abstracts of relevant literature from the databases until January 18, 2023. Two investigators (K.T. and P.P.) independently screened records for relevant evidence using Rayyan, a web application for screening systematic review records [24]. Any disagreements during the screening procedures were resolved by consulting with an expert in cardiac electrophysiology (K.S.).

## Data extraction

Two authors (K.T. and P.P.) extracted the information as follow: study authorship, year of publication, study period, country/location, study design, patient demographics including mean age, sample size, number of people diagnosed with DM, definitions of DM, types (or sizes) of particulate matter, type (or length) of exposure (i.e., short- or long-term), outcomes measurements/definitions (i.e., hospital admission, or ER visit), and association parameters reported (i.e., odds ratio (OR), relative risk (RR), or hazard ratio (HR)). Any disagreement during the process was resolved by discussion with the cardiac electrophysiology expert (K.S.). For studies with incomplete data required for the analysis, we contacted the authors of those studies via email. If they did not respond within 2 weeks, the data would be imputed or reported as missing. WebPlotDigitizer was used to extract numeric data from graphs where appropriate.

## Risk of bias assessment

Two authors (K.T. and P.P.) autonomously checked the quality of each study included using the Newcastle-Ottawa Scale (NOS) quality assessment tool. NOS assessed the quality of study

and bias based on three separate domains, which are subject selection, comparability, and out-come assessment. In accordance with the number of stars received in each domain, studies will be appraised as good, fair, or poor quality. Studies with 3 to 4 stars in the selection domain, 1 to 2 stars in the comparability domain, and 2 to 3 stars in the outcome domain will be appraised as good quality. Studies with 2 stars in the selection domain, 1 to 2 stars in the com-parability domain, and 2 to 3 stars in the outcome domain will be appraised as fair quality. Studies with 0 to 1 star in the selection domain, or 0 stars in the comparability domain, or 0 to 1 star in the outcome domain will be appraised as poor quality. Any disagreements were resolved during the quality assessment process by consulting with the expert in cardiac electro-physiology (K.S.) and the clinical methodologist (P.P.).

## Synthesis method

All analyses were conducted using Stata 17 (StataCorp, College Station, Texas, USA). P-values <0.05 was regarded statistically significant. A traditional approach of a pairwise meta-analysis was used for data synthesis. The statistical heterogeneity will be assessed by the use of the Cochrane Q test and I-squared statistics. As all studies are expected to exhibit clinical and methodological heterogeneity, we used the DerSimonian and Laird (DL) random-effects model to estimate the pooled OR with its 95% confidence intervals (CIs). The decision to use the DL random-effects model was not based on any threshold of the heterogeneity statistics. Studies that did not report the ORs or did not provide adequate data to calculate the ORs were not included in data synthesis. For our analysis, we defined exposure status as follows: expo-sure to PM of any size (combining both PM2.5 and PM10), exposure to PM2.5, and exposure to PM10. In the case of exposure to PM of any size, we would identify the effects of both PM2.5 and PM10. Considering that people in real environments are exposed to a mixture of the two, identifying PM in general, regardless of size, may be more relevant and better inform public health policy.

We performed subgroup analysis based on diabetic status to investigate whether diabetic status modifies the effect of the association between PM exposures and cardiac arrhythmias. Additionally, subgroup analysis was performed for the overall population based on the risk of bias in each study. Publication bias would be assessed with a funnel plot and Egger's test if the number of included studies were higher or equal to ten. In the case where publication bias is likely, the Trim-and-fill method would be used to assess and adjust the pooled results for pub-lication bias.

We conducted a post hoc sensitivity analysis to examine the robustness of our pooled results in cases where the estimation of heterogeneity parameters might be unreliable due to the low number of studies included in the random-effects model [25]. This was done by per-forming all analyses using both the Hartung-Knapp-Sidak-Jonkman (HKSJ) method for the random-effects model and the fixed-effect model as suggested in literature [26, 27]. Subse-quently, we compared the pooled results obtained from the three methods. If all the results were consistent in terms of statistical significance, then our primary analyses using the DL ran-dom-effects model could be considered robust.

## Results

### Study selection

The search from databases and citations found 215 records, and 44 duplications were removed. No additional study was identified from the grey literature search. The titles and abstracts were screened, and 159 records were excluded. We screened the full text of the 12 remaining records, with 1 of them being excluded due to a study design that was not suitable for our

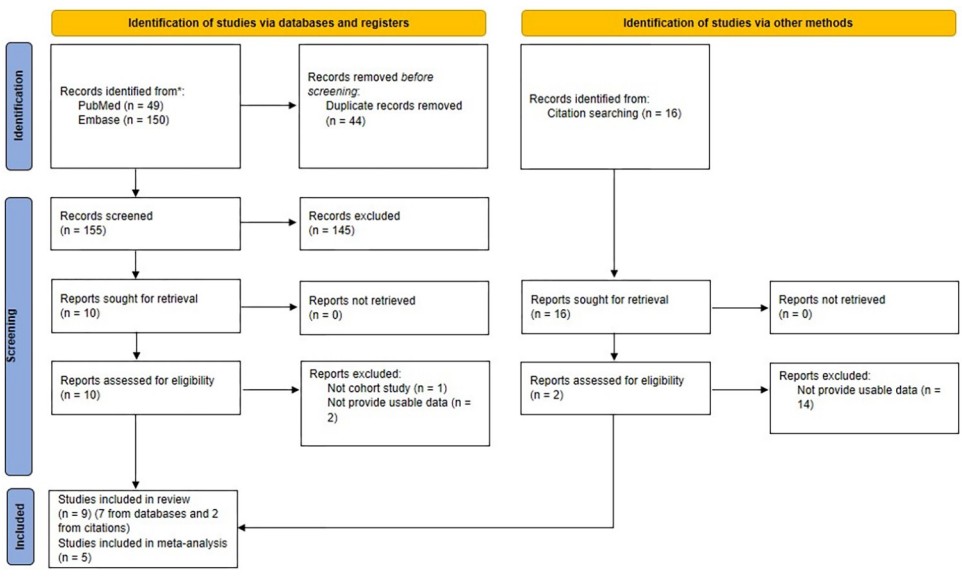

**Fig 1. PRISMA flow diagram.**

inclusion criteria, and 2 of them did not provide usable data. Nine remaining studies were included in our systematic review [28–36], and five of them were included in our meta-analysis (Fig 1) [28, 29, 32, 34, 35]. All the data required for analysis were reported or visualized, and no imputation of data was made during the extraction process. The association parameters for the following studies were digitized from graphs: Jin (2022) [30], Lee (2019) [32], Liang (2021) [33], Zanobetti (2014) [35], and Zheng (2018) [36].

## Study characteristics

The systematic review included a total of 9 studies. Two studies were conducted in Taiwan, two in the USA [30, 35], two in China [33, 36], one in Georgia [34], one in South Korea [31], and one in Sweden [29]. Five studies were case-crossover studies [28, 32–34, 36], and 4 studies were cohort studies [29–31, 35]. Four studies focused on the effect of PM2.5 [30, 32, 33, 35], two on the effect of PM10 [28, 34], and three on both PM2.5 and PM10 [29, 31, 36]. Seven studies focused on the short-term effect of PM [28, 29, 32–36], and two on the long-term effect [30, 31]. Most studies did not provide detailed data on DM diagnostic criteria or information on DM types. Three studies defined DM based on ICD-9 and ICD-10 codes [28, 34, 36]. The summarized characteristics of the included studies are presented in Table 1.

Seven studies investigated the impact of PM2.5 on cardiac arrhythmias and reported subgroup analysis results based on diabetic status [29–33, 35, 36]. Among these, six studies demonstrated an association between PM2.5, DM, and cardiac arrhythmias. Notably, the studies by Dahlquist et al. and Kim et al. were exceptions, as they indicated that DM did not modify the effects of PM exposure on arrhythmias [29, 31]. Additionally, five studies focused on the effect of PM10 on cardiac arrhythmias, with subgroup analyses based on diabetic status conducted [28, 29, 31, 34, 36]. Among these, three studies revealed an association between PM10, DM, and cardiac arrhythmias. Conversely, the study by Chiu and colleagues suggested that DM did not modify the effects of PM exposure on arrhythmias [28].

According to the five studies included in our meta-analysis [28, 29, 32, 34, 35], a total of 4,431,452 participants were included, with 2,556 of them being diabetic patients. The average age of participants was 74 years old. The studies employed various outcome measurements,

**Table 1. Summarized characteristics of the included studies.**

| Author (year) | Period | Country/ location | Study design | Type (duration) of effect | Total sample size | N (DM) | Mean age (year) | Diabetes definition | Outcome measurement/ definition | PM | Parameter used | Outcome for DM group | Outcome for non-DM group |
|---|---|---|---|---|---|---|---|---|---|---|---|---|---|
| Peel (2007) [34] | 1993–2000 | Georgia | Case-crossover | Short term | 27,342 | 1,562 | NR | ICD9 code 250; type II or unspecified type | ER visits due to dysarrhythmia (ICD 9 code 427) | 10 | OR for an increase in 1 SD of PM | OR = 1.049 (0.968, 1.137) | OR = 1.0109 (0.989, 1.029) |
| Chiu (2009) [28] | 2000–2006 | Taiwan | Case-crossover | Short term | 21,581 | 452 | NR | ICD9 code 250; type II or unspecified type | Hospital admission due to dysarrhythmia (ICD 9 code 427) | 10 | OR for an increase in 1 IQR of PM | OR = 0.95 (0.56, 1.63) | OR = 1.08 (1.04, 1.13) |
| Zanobetti (2014) [35] | 2000–2010 | USA | Cohort | Short term | 1,448 | 305 | 75.3 | Physician diagnosis of diabetes, use of any diabetes medications, or fasting glucose >126 mg/dL | Arrhythmia episodes measured as ventricular ectopy on EKG | 2.5 | OR for an increase in 1 IQR of PM | OR = 1.54 (1.06, 2.4) | OR = 1.34 (1.16, 1.64) |
| Zheng (2018) [36] | 2014–2015 | China | Case-crossover | Short term | 175,265 | 19,947 | NR | ICD10 E10 to E14, E10 = T1DM, E11 = T2DM, E12 = malnutrition-related DM, E13 = other specified DM, and E14 = unspecified DM | Hospital admission due to arrhythmia (ICD10 codes: I44, I45, I46, I47, I48, I49) | 2.5 / 10 | Percent change per 1 IQR increase of PM | Percent change = 4.93 (2.51, 7.34) / Percent change = 7.16 (4.03, 10.30) | Percent change = 1.88 (1.07, 2.78) / Percent change = 2.69 (1.52, 3.85) |
| Kim (2019) [31] | 2009–2013 | South Korea | Cohort | Long term | 432,587 | 27,627 | NR | Diabetes mellitus (no additional specification on diagnosis criteria, coding used, or types of DM) | Outpatient hospital visits or hospital admissions with atrial fibrillation (AF) as likely diagnosis | 2.5 / 10 | HR for every 10 μg/m$^3$ increase in PM | HR = 1.19 (1.18, 1.21) / NR | HR = 1.18 (1.18, 1.19) / NR |
| Lee (2019) [32] | 2006–2011 | Taiwan | Case-crossover | Short term | 670 | 213 | 70.5 | Diabetes mellitus (no additional specification on diagnosis criteria, coding used, or types of DM) | Hospital admission with first diagnosis of atrial fibrillation (AF) based on ICD10 code 437.31 | 2.5 | OR for an increase in 1 IQR of PM | OR = 1.34 (1.02, 1.86) | OR = 1.20 (0.81, 1.28) |
| Dahlquist (2020) [29] | 2012–2013 and 2016–2018 | Sweden | Cohort | Short term | 218 | 24 | NR | Diabetes mellitus (no additional specification on diagnosis criteria, coding used, or types of DM) | EKG measurement suggests atrial fibrillation (AF) | 2.5 / 10 | OR for an increase in 1 IQR of PM | OR = 0.98 (0.78, 1.24) / OR = 1.48 (1.14, 1.92) | OR = 1.06 (0.98, 1.15) / OR = 1.08 (0.99, 1.18) |
| Liang (2021) [33] | 2011–2014 | China | Case-crossover | Short term | 8,241 | NR | 60 | Diabetes mellitus (no additional specification on diagnosis criteria, coding used, or types of DM) | ER visits with first diagnosis of atrial fibrillation (AF) based on ICD9 code, confirmed by electrocardiogram | 2.5 | RR for each cut point of PM concentration | RR was reported on graph | RR was reported on graph |

*(Continued)*

**Table 1.** (Continued)

| Author (year) | Period | Country/ location | Study design | Type (duration) of effect | Total sample size | N (DM) | Mean age (year) | Diabetes definition | Outcome measurement/ definition | PM | Parameter used | Outcome for DM group | Outcome for non-DM group |
|---|---|---|---|---|---|---|---|---|---|---|---|---|---|
| Jin (2022) [30] | 2000–2016 | USA | Cohort | Long term | 32,165,933 | NR | NR | Diabetes mellitus (no additional specification on diagnosis criteria, coding used, or types of DM) | First occurrence of atrial fibrillation (AF) identified using an algorithm that incorporates information across all available medicare claims databases | 2.5 | HR for every 1 μg/m$^3$ increase in PM | HR = 1.010 (1.008, 1.011) | HR = 1.003 (1.002, 1.004) |

**Abbreviations:** DM, diabetes mellitus; EKG, electrocardiogram; HR, hazard ratio; ICD, International Classification of Diseases; IQR, interquartile range; NR, not reported; OR, odds ratio; PM, particulate matter; RR, relative risk; SD, standard deviation; T1DM, type 1 diabetes mellitus; T2DM, type 2 diabetes mellitus; USA, United States of America.

**Table 2. Risk of bias.**

| Author (year) | Representativeness of the exposed cohort | Selection of the non-exposed cohort | Ascertainment of exposure | Demonstration that outcome of interest was not present at start of study | Comparability | Assessment of outcome | Was follow-up long enough for outcomes to occur | Adequacy of follow up of cohorts | Total score | Quality of study |
|---|---|---|---|---|---|---|---|---|---|---|
| Peel (2007) | * | * | * |  | * | * | * |  | 6 | Good |
| Chiu (2009) | * | * | * | * | * | * | * |  | 7 | Good |
| Zanobetti (2014) |  |  | * | * |  | * | * | * | 5 | Poor |
| Zheng (2018) | * | * | * |  | * | * | * |  | 6 | Good |
| Kim (2019) | * | * | * | * |  | * | * |  | 6 | Fair |
| Lee (2019) | * | * | * | * | * | * | * |  | 7 | Good |
| Dahlquist (2020) | * | * | * | * | * |  |  | * | 6 | Fair |
| Liang (2021) | * | * | * |  | * | * |  |  | 5 | Poor |
| Jin (2022) | * | * | * | * |  | * | * |  | 6 | Fair |

with two focusing on hospital admissions for cardiac arrhythmias [28, 32, 36], two utilizing ECG measurements [29, 35], and one centering on emergency room visits for cardiac arrhythmias [34]. All five studies focused on short-term effects of PM exposure.

## Risk of bias in studies

Out of the nine studies included in our systematic review, four were deemed to be of good quality [28, 32, 34, 36], three of fair quality [29–31], and two of poor quality [33, 35]. There appears to be a tendency for bias in the comparability of the studies, which could potentially impact the outcome of the meta-analysis. When focusing solely on the studies included in the meta-analysis, three were rated as good [28, 32, 34], one as fair [29], and one as poor in quality [35]. The risk of bias assessments is presented in Table 2.

## Exposure to PM and cardiac arrhythmias

Of the five PM studies included, they were categorized based on types of PM into twelve sub-studies and included in the overall analysis. Short-term exposure to PM of any size was associated with the occurrence of cardiac arrhythmia events leading to hospital or emergency visits in the overall population (OR 1.10 (95% CI 1.01, 1.38), $p < 0.001$). This meta-analysis revealed that the modifying effect of DM status was not statistically significant (OR of DM group 1.08 (95% CI 1.02, 1.14), $p < 0.001$, and OR of non-DM group 1.18 (95% CI 1.01, 1.38), $p = 0.041$, p for group difference = 0.304), but the DM group tended to have a larger effect size than the non-DM group (Fig 2).

## Exposure to PM2.5 and cardiac arrhythmias

Three studies with 2,336 participants focused on PM2.5 and were included and divided into six sub-studies. The meta-analytic results showed a significant effect of exposure to PM2.5 on cardiac arrhythmias in the overall population (OR 1.18 (95% CI 1.04, 1.34), $p = 0.008$).

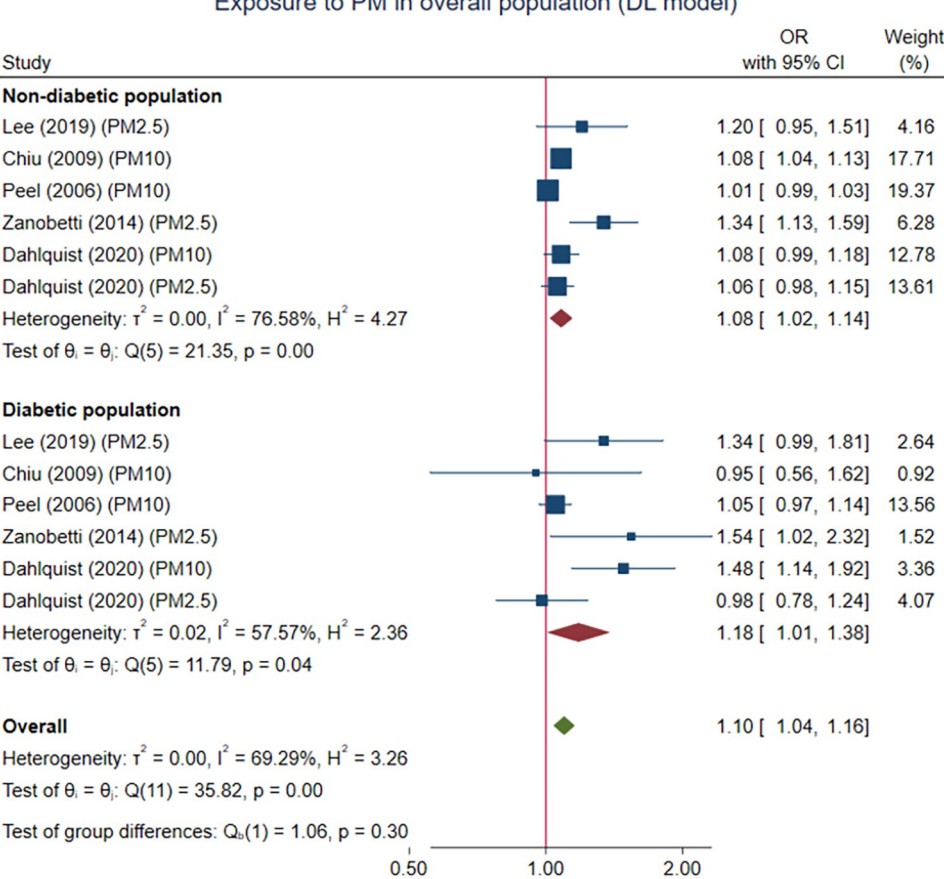

**Fig 2. The pooled association between PM exposure and cardiac arrhythmias in the overall population with a subgroup analysis based on diabetic status.** The effect parameters are expressed as odds ratio (OR) of higher PM exposure over lower PM exposure.

Exposure to PM2.5 demonstrated a significant effect on cardiac arrhythmias in the non-DM group (OR 1.18 (95% CI 1.00, 1.38), p = 0.045), but the effect on cardiac arrhythmias in the DM group was not statistically significant (OR 1.22 (95% CI 0.93, 1.60), p = 0.092). The statistical evidence to support the modifying effect of DM was not statistically significant (p = 0.813), but the results still showed a trend of a larger OR in the DM group than the non-DM group (Fig 3).

## Exposure to PM10 and cardiac arrhythmias

Three studies (six sub-studies) with 4,429,334 participants focused on PM10 and were included. A significant effect of exposure to PM10 on cardiac arrhythmias in the overall population was observed (OR 1.06 (95% CI 1.01, 1.12), p = 0.028). The effect of PM10 exposure on cardiac arrhythmias in both the DM group (OR 1.05 (95% CI 0.99, 1.11), p = 0.090) and non-DM group was not statistically significant (OR 1.16 (95% CI 0.89, 1.51), p = 0.261). DM status did not significantly modify the effect between PM10 and the occurrence of cardiac arrhythmias (p = 0.455) but showed a trend of a larger effect size in the DM group than the non-DM group (Fig 4).

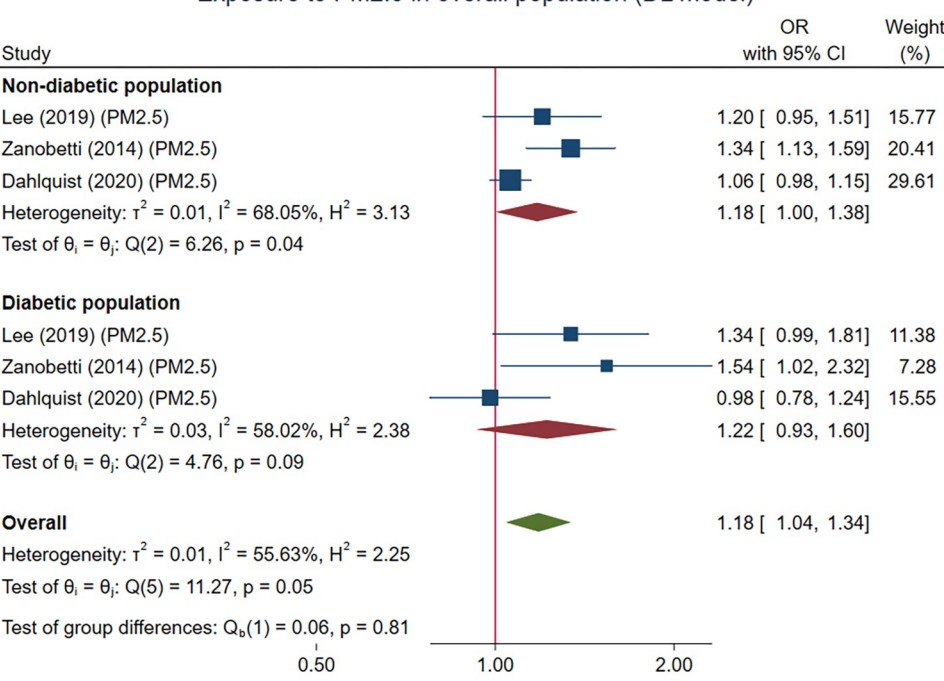

**Fig 3. The pooled association between PM2.5 exposure and cardiac arrhythmias in the overall population with a subgroup analysis based on diabetic status.** The effect parameters are expressed as odds ratio (OR) of higher PM2.5 exposure over lower PM2.5 exposure.

## The effect of study quality on the association between PM exposure and cardiac arrhythmias

The quality of each study affected the association between exposure to PM and cardiac arrhythmia events in the overall population. We observed a higher value of pooled OR in studies with poor quality (OR 1.37 (95% CI 1.17, 1.61), $p < 0.001$), followed by studies with fair (OR 1.10 (95% CI 0.99, 1.21), $p = 0.067$) and good quality (OR 1.06 (95% CI 1.00, 1.11), $p = 0.043$). The test for subgroup difference according to study quality showed statistical significance ($p = 0.010$) (Fig 5).

## Publication bias

We were able to assess the presence of publication bias only for the pooled association between PM and cardiac arrhythmias in the overall population. This was the only analysis with a total number of studies above ten. The funnel plot of this analysis shows significant asymmetry with potential small study effects on the left side of the plot (Fig 6A). The Egger's test also revealed a statistically significant result at $p = 0.002$, representing evidence for publication bias. To account for this bias, we conducted a Trim-and-fill method to examine the influence of publication bias on our previously estimated pooled OR, which was 1.10 (95%CI 1.04, 1.16). The pooled OR, after accounting for publication bias, was estimated at 1.08 (95%CI 1.02, 1.14) (Fig 6B).

## Post-hoc sensitivity analysis

It was found that the effects of PM of any size were significant in the overall population for all three analytic models (Table 3). The association between PM of any size was also significantly

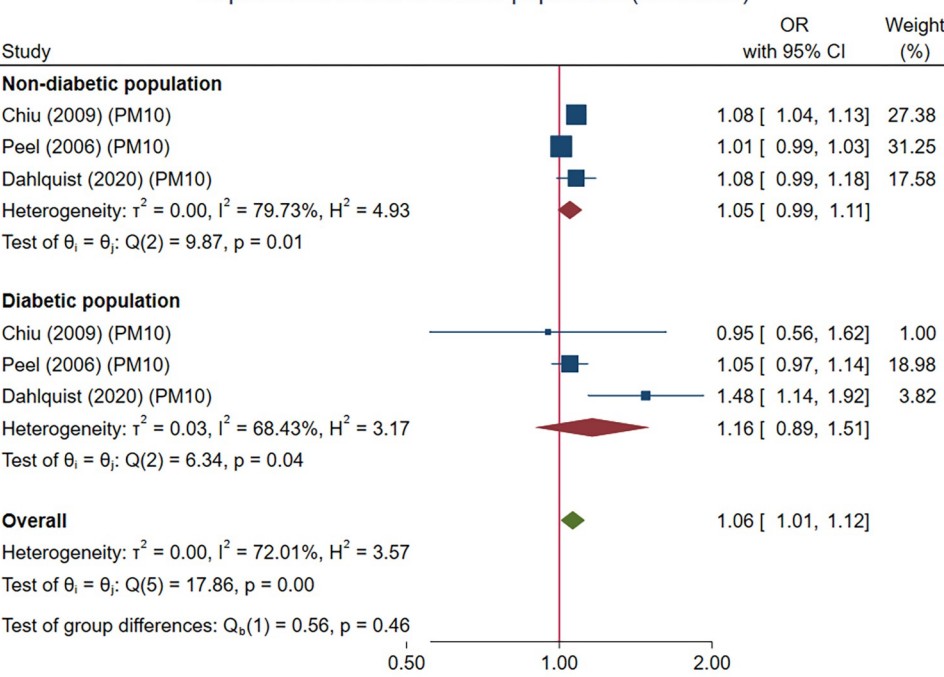

**Fig 4. The pooled association between PM10 exposure and cardiac arrhythmias in the overall population with a subgroup analysis based on diabetic status.** The effect parameters are expressed as odds ratio (OR) of higher PM10 exposure over lower PM10 exposure.

associated with arrhythmia in both non-diabetic and diabetic subgroups when the DL random-effects model was used (Fig 2). When the HKSJ method was used, the results were similar in terms of the effect estimates but not in terms of statistical significance (S1-S3 Figs in S2 File). The fixed-effect model provides statistically significant results for both the overall and the subgroup analysis (S4 Fig in S2 File). However, the pooled ORs were quite different from those of random-effects models used. For PM2.5, all three modeling approaches provide consistent results for the effect of PM2.5 and arrhythmia in the overall population (S5 Fig in S2 File), but not in the subgroup by DM status (S6, S7 Figs in S2 File). For PM10, the associations identified from all three modeling techniques were not consistent in both the overall and subgroup by diabetic status (S9-S12 Figs in S2 File).

## Discussion

Based on the gathered studies on PM and cardiac arrhythmia, considering diabetes as a potential modifier, our results suggest that short-term exposure to higher concentrations of PM is linked to increased hospital or emergency visits due to cardiac arrhythmias. However, it appears that the overall effect of PM is primarily driven by exposure to higher concentrations of PM2.5 rather than PM10. The effects identified in our analysis were not modified by the diabetic status of the population. Nevertheless, there was a tendency for the magnitude of the effect to be more pronounced among the diabetic population than the non-diabetic population.

To date, the underlying mechanisms of the association between PM and cardiac arrhythmias in diabetic patients are still ambiguous. In 2010, a study by Schneider and colleagues found that exposure to PM2.5 caused systemic inflammation and altered ventricular

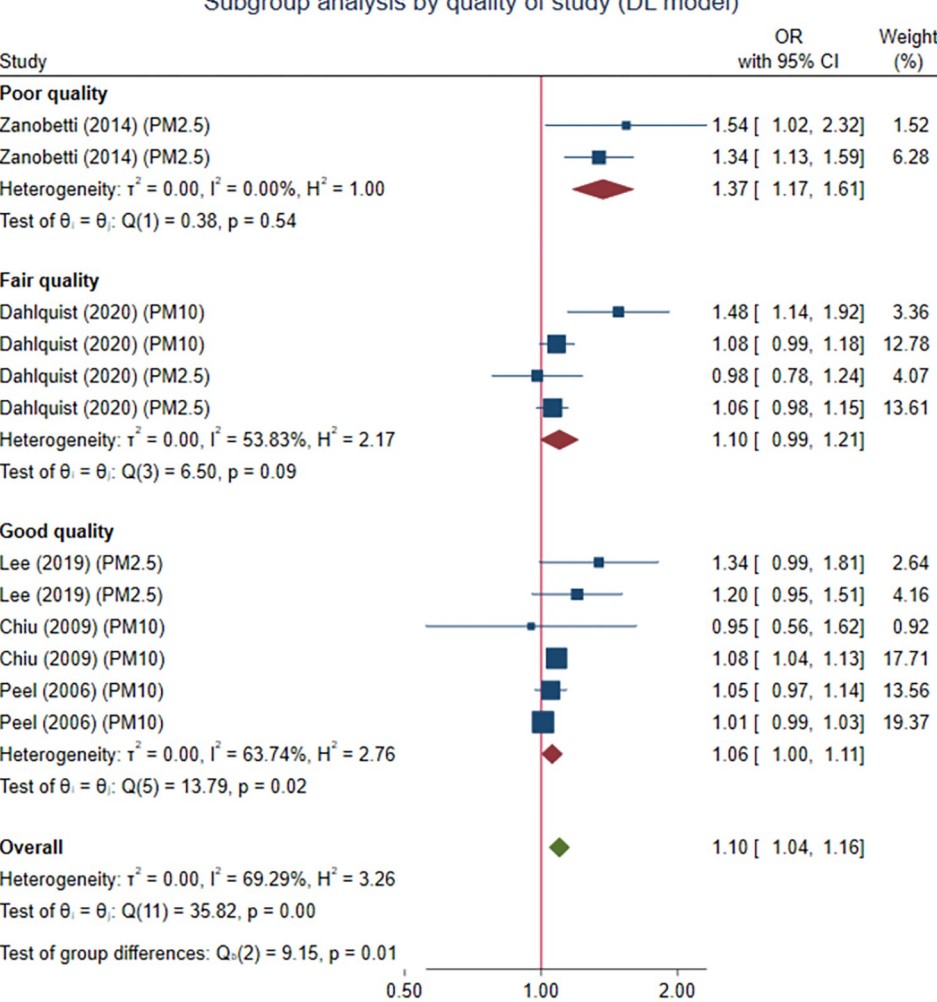

**Fig 5. Subgroup analysis based on study quality of the pooled association between PM exposure and cardiac arrhythmias in the overall population.** The effect parameters are expressed as odds ratio (OR) of higher PM exposure over lower PM exposure.

repolarization in diabetic individuals, which might lead to cardiac arrhythmias [17]. However, their findings were based on the observed changes in cardiac rhythm and electrocardiographic parameters, not on the actual documented clinical events. Moreover, they included a rather small sample size of diabetic individuals without comparing the results to non-diabetic controls. There were several studies that reported the significant association between PM and documented hospital or emergency visits due to cardiac arrhythmias in the general population, some of which were included in this review. Nonetheless, most of them focused on a particular type of arrhythmia, such as AF, and did not examine the potential effect modification by performing a subgroup analysis based on diabetic status [18–22]. In this review, we aimed to identify all studies that provided data on the effect of PM exposure on cardiac arrhythmia events in both diabetic and non-diabetic subgroups of the population to enable us to examine the potential effect modification.

Of the nine studies included in our review, most concluded that diabetic patients are potentially more susceptible to the effects of PM on arrhythmic events [x]. It is important to note

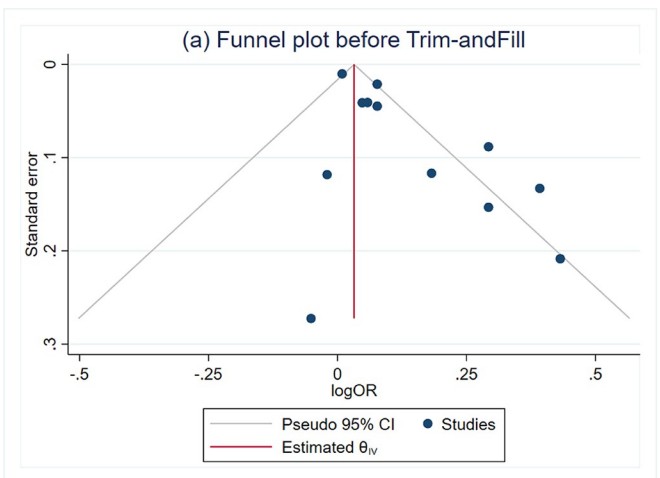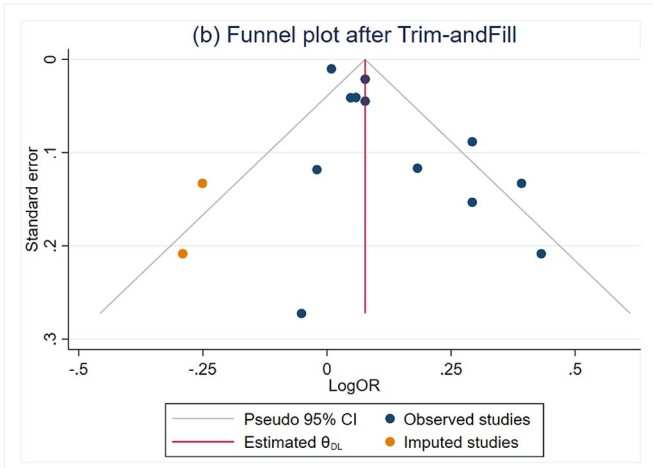

**Fig 6.** Funnel plots before (a) and after Trim and Fill method (b) for the meta-analysis of PM exposure and cardiac arrhythmias in the overall population.

that not all studies identified a statistically significant interaction, and some conclusions might be based solely on a seemingly higher effect size in the diabetic group. Two studies did not suggest that the effect of PM exposure was modified by DM status, namely the studies by Chiu et al. [28] and Kim et al. [31]. In their analysis, Kim et al. performed a subgroup analysis by DM status and did not identify a statistically significant interaction for exposure to PM2.5 and AF incidence [31]. However, the OR in the DM group was still higher than in the non-DM group, with p for interaction close to the significance threshold. In contrast, Chiu et al. did not identify any evidence of effect modification by DM status, and the OR in the DM group was even smaller than in the non-DM group. They proposed that their inverse findings might be due to chance [28]. When we compared the findings of Chiu et al. [28] to those of Peel et al. [34], the most similar study in terms of design and analysis, we hypothesized that the

**Table 3. Results of the post-hoc sensitivity analysis comparing the pooled odds ratio (ORs) across three meta-analytic models.**

|  | DL random-effects analysis | HKSJ random-effects analysis | Fixed-effect analysis |
|---|---|---|---|
|  | Pooled OR (95%CI) | Pooled OR (95%CI) | Pooled OR (95%CI) |
| **PM (any size)** |  |  |  |
| Overall (n = 12) | **1.10 (1.04, 1.16)** * | **1.13 (1.03, 1.23)** * | **1.03 (1.02, 1.05)** * |
| Non-diabetic (n = 6) | **1.08 (1.02, 1.14)** * | 1.09 (0.99, 1.20) | **1.03 (1.01, 1.05)** * |
| Diabetic (n = 6) | **1.18 (1.01, 1.38)** * | 1.19 (0.96, 1.46) | **1.09 (1.02, 1.17)** * |
| **Only PM2.5** |  |  |  |
| Overall (n = 6) | **1.18 (1.04, 1.34)** * | **1.19 (1.01, 1.39)** * | **1.12 (1.05, 1.19)** * |
| Non-diabetic (n = 3) | **1.18 (1.00, 1.38)** * | 1.17 (0.86, 1.59) | **1.11 (1.04, 1.19)** * |
| Diabetic (n = 3) | 1.22 (0.93, 1.60) | 1.22 (0.68, 2.18) | 1.17 (0.99, 1.38) |
| **Only PM10** |  |  |  |
| Overall (n = 6) | **1.06 (1.01, 1.12)** * | 1.08 (0.96, 1.23) | **1.03 (1.01, 1.04)** * |
| Non-diabetic (n = 3) | 1.05 (0.99, 1.11) | 1.05 (0.94, 1.16) | **1.02 (1.01, 1.04)** * |
| Diabetic (n = 3) | 1.16 (0.89, 1.51) | 1.16 (0.68, 1.99) | 1.08 (1.00, 1.16) |

**Abbreviations:** CI, confidence interval; DL, DerSimonian-Laird; HKSJ, Hartung-Knapp-Sidak-Jonkman; OR, odds ratio; PM, particulate matter. **Note:**

* signifies statistical significance at p-value less than 0.05

difference in results might be explained by variations in geographical region, race, outcome definitions, and parameters used during analysis.

According to our meta-analysis of five studies, we found a significant association between short-term exposure to higher PM concentrations and hospital or emergency visits due to cardiac arrhythmias in the general population. When subgroup analysis by PM types was performed, we observed that the occurrence of cardiac arrhythmia events might be more influenced by exposure to PM2.5 than PM10. The effect of PM10 and arrhythmia events identified in the DL random-effects model was significant in the overall population, but when the HKSJ random-effects model was used, statistical significance was not identified. The direction of effect for both types of PM was still consistent and agreeable with prior studies. The study by Yue and colleagues showed that PM2.5 and PM10 were associated with atrial fibrillation episodes, whether short or long-term exposure. Although the results were in the same direction, it focused only on atrial fibrillation, which is one type of cardiac arrhythmia [37]. The study by Song and colleagues showed that both PM2.5 and PM10 were associated with hospitalization and mortality due to cardiac arrhythmias, which is consistent with our results [18].

From subgroup analysis by DM status, the effects of PM exposure tended to be higher in the diabetic than the non-diabetic population for all three exposure definitions of PM. However, no statistical significance was identified. Thus, based on our findings here, we conclude that there is still insufficient evidence to determine that diabetic status modifies the effect of PM exposure on cardiac arrhythmia events requiring emergency or hospital visits. The results align with the prior systematic review and meta-analysis. The study by Yue and colleagues indicated in the subgroup analysis that diabetic status did not show an effect on the association between PM and atrial fibrillation [37]. Although our findings did not support our prior hypothesis, we would not say that the findings were against or contrary to what we would expect. The main reason is that higher odds of arrhythmia events were still observed in diabetic groups for all analyses, but the differences were quite small and might not be adequately powered to identify significant interactions. Moreover, it is important to note that our meta-analysis consisted of a subset of studies that reported effect parameters as ORs, and most studies included in this review still provide the conclusion that DM should be regarded as a susceptible group to the effect of PM exposure on the occurrence of arrhythmias.

Cardiac arrhythmias are multifactorial, meaning that factors other than DM can contribute to the occurrence of symptoms and exhibit a confounding effect. The main issue encountered in this review is the comparability of the included evidence. First, some studies indicated that their results were associated with PM, but there was no evidence that the effect of other pollutants was adjusted. This may lead to overestimation in some studies. Second, most participants in all studies had comorbidities or conditions such as hypertension or congestive heart failure. When subgroup analysis of diabetic status was done, the non-DM cohort might be patients with comorbidities, rather than healthy subjects, which might show the confounding effect, making the effect of diabetic status underestimated. Further study with a low risk of bias in the comparability domain is required. Medications used by DM patients could be another possible reason for the irrelevant result. Some studies indicated that certain antidiabetic drugs showed a protective effect against cardiac arrhythmias. The network meta-analysis conducted by Shi and colleagues showed that antidiabetic drugs could lower the risk of AF in DM patients, and glucagon-like peptide-1 receptor agonists were the most effective among the medications [38]. The nationwide cohort study in South Korea conducted by Kim and colleagues showed that metformin and thiazolidinediones could prevent AF in DM patients [39].

There were some limitations in our review. In terms of the overall characteristics of the included studies, significant demographic and methodological heterogeneity was observed. The definitions of DM and arrhythmia varied, and the interpretation of results should be

made carefully. In addition, the quality of the study according to the risk of bias was mixed, and there was a trend observed that studies with a higher risk of bias seemed to overestimate the effect of PM compared to studies with a lower risk of bias.

For the review and synthesis process, there were several points to be addressed. Firstly, only studies that reported the effect of PM on cardiac arrhythmias for both DM and non-DM were included in our review, so the results regarding the effect of PM on cardiac arrhythmias were synthesized from a smaller number of studies than they should have been. However, this was the only appropriate approach to examine the effect modification of DM on the association of interest. Secondly, the evidence for publication bias was quite significant, reflecting the tendency of studies with a positive association between PM and arrhythmia to be published. Nonetheless, the Trim-and-Fill method indicated that the effect of bias might be minuscule. Thirdly, performing a meta-analysis using a small number of studies threatens the validity of the statistical model, especially when the random-effects model was pre-specified. Alternative modeling methods were used to examine the robustness of the primary results and found that only a few aspects of the analyses were robust. Fourthly, due to the scarcity of the data reported in original articles, we could not perform the analysis to examine the differential effects for each type of DM. Finally, our analytic approach was univariable, and thus, confounding was not adjusted. In an observational context, it is very likely that the results might be influenced by other factors, both at the environmental or individual level. Future studies should properly adjust the confounding effect, as cardiac arrhythmias are multifactorial. Gathering and combining individual patient data (IPD) that has complete information on all potential confounders, including comorbidities, medications used, and family history of cardiac arrhythmias, to perform a more sophisticated analysis of large observational research or conduct an IPD meta-analysis may result in more valid answers to the question.

## Conclusion

Our systematic review and meta-analysis revealed that short-term exposure to higher PM concentrations, specifically PM2.5, may have a significant effect on the occurrence of cardiac arrhythmia requiring hospital or emergency visits. Although the test for diabetic status as an effect modifier was not statistically significant, there was a tendency that the effect of PM exposure was modestly higher in the diabetic than the non-diabetic population. We suggest that diabetic patients should still be considered as a susceptible group to the effect of PM exposure on cardiac arrhythmia events. Future studies with a large sample size and low risk of bias, especially in the comparability or confounding domain, are warranted.

## Supporting information

**S1 Checklist. PRISMA 2020 checklist.**
(DOCX)

**S1 File. PROSPERO protocol registration.**
(PDF)

**S2 File.**
(DOCX)

## Author Contributions

**Conceptualization:** Kiattichat Tassanaviroj, Pimchanok Plodpai, Krittai Tanasombatkul, Krekwit Shinlapawittayatorn, Phichayut Phinyo.

**Data curation:** Kiattichat Tassanaviroj, Pimchanok Plodpai, Krekwit Shinlapawittayatorn, Phichayut Phinyo.

**Formal analysis:** Kiattichat Tassanaviroj, Pakpoom Wongyikul, Krittai Tanasombatkul, Phichayut Phinyo.

**Investigation:** Kiattichat Tassanaviroj, Pakpoom Wongyikul, Krittai Tanasombatkul, Krekwit Shinlapawittayatorn, Phichayut Phinyo.

**Methodology:** Kiattichat Tassanaviroj, Pimchanok Plodpai, Pakpoom Wongyikul, Krittai Tanasombatkul, Krekwit Shinlapawittayatorn, Phichayut Phinyo.

**Project administration:** Phichayut Phinyo.

**Resources:** Krittai Tanasombatkul.

**Software:** Phichayut Phinyo.

**Supervision:** Phichayut Phinyo.

**Validation:** Krekwit Shinlapawittayatorn, Phichayut Phinyo.

**Writing – original draft:** Kiattichat Tassanaviroj.

**Writing – review & editing:** Kiattichat Tassanaviroj, Pimchanok Plodpai, Pakpoom Wongyikul, Krittai Tanasombatkul, Krekwit Shinlapawittayatorn, Phichayut Phinyo.

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
