## [Decision Letter · Decision Letter 0]

16 Jan 2024

PONE-D-23-26753Effect modification of diabetic status on the association between exposure to particulate matter and cardiac arrhythmias: a systematic review and meta-analysisPLOS ONE

Dear Dr. Phinyo,

Thank you for submitting your manuscript to PLOS ONE. After careful consideration, we feel that it has merit but does not fully meet PLOS ONE’s publication criteria as it currently stands. Therefore, we invite you to submit a revised version of the manuscript that addresses the points raised during the review process.

We look forward to receiving your revised manuscript.

Kind regards,

Trenton Honda

Academic Editor

PLOS ONE

Journal Requirements:

Reviewers' comments:

Reviewer's Responses to Questions

**Comments to the Author**

1. Is the manuscript technically sound, and do the data support the conclusions?

Reviewer #1: Yes

Reviewer #2: Partly

2. Has the statistical analysis been performed appropriately and rigorously? 

Reviewer #1: Yes

Reviewer #2: I Don't Know

3. Have the authors made all data underlying the findings in their manuscript fully available?

Reviewer #1: No

Reviewer #2: Yes

4. Is the manuscript presented in an intelligible fashion and written in standard English?

Reviewer #1: No

Reviewer #2: Yes

5. Review Comments to the Author

Reviewer #1: Please see attached review notes.

Overall comments:

• Overall, the methodological approach and execution of the systematic review and meta-analysis appear sound.

• Good use of Cochrane guidelines and pre-registration of the study.

• The manuscript would benefit from English language copyediting to correct some issues with grammar and flow.

• The small number of studies for meta-analysis (3 each for PM2.5 and PM10) calls into question the use of random effects model and whether it is sufficiently powered; recommend strengthening rationale for use of this model and explanation of the power expectations. (Although I2 is appropriately used to assess heterogeneity, please note the Cochrane caution, “Authors should recognize that there is much uncertainty in measures such as I2 and Tau2 when there are few studies. Thus, use of simple thresholds to diagnose heterogeneity should be avoided.”) [Chapter 10: Analysing data and undertaking meta-analyses | Cochrane Training]. Consider whether the number of studies is sufficient to justify the meta-analysis or if it should be dropped (for example, Chiu’s 1% weight in PM10 analysis is a vanishingly small contribution).

• The reporting of results is generally good, but when reporting the effects of both PM10 and PM2.5 on arrythmias, sometimes the language is unclear. Please specify how particulate exposure was combined when reporting these together. Furthermore, the manuscript would be strengthened by explaining why combining PM10 and PM2.5 results makes sense in the first place.

• Publication bias is a notorious potential issue in meta-analysis; although the authors discuss searching for “grey literature,” it is not clear whether such studies were uncovered and how they were handled to eliminate bias. A funnel plot would be useful to elucidate.

• The manuscript would benefit from a discussion of type 1 & 2 DM, whether they play differing roles in PM effects, and how this is reflected (or deemed irrelevant) in these findings.

• Methodologically, the use of imputation and final missingness are an important details that receives no detailed description.

Reviewer #2: Abstract, title and references

● Is the aim clear?

● Is it clear what the study found and how they did it?

● Is the title informative and relevant?

● Are the references:

● Relevant?

● Recent?

● Referenced correctly?

● Are appropriate key studies included?

The aim is clear, however, it may be ideal to add to the population the study is targeting. The same applies to the title.

It is clearly stated what was found in the study.

The references are relevant, recent, and used correctly.

Introduction/background

● Is it clear what is already known about this topic?

● Is the research question clearly outlined?

● Is the research question justified given what is already known about

the topic?

It is clear what is already known about the topic and the research question is clearly

outlined and justified given the limited information on what is already known about

the topic.

However, I find the introduction too short. The authors discuss about oxidative stress and inflammation but this has little to do with the article itself. The authors should discuss more about PM, its association with arrhythmias and which arrhythmias in particular, the impact of diabetes on arrhythmias, the impact of PM on diabetes etc.

Methods

● Is the process of subject selection clear?

● Are the variables defined and measured appropriately?

● Are the study methods valid and reliable?

● Is there enough detail in order to replicate the study?

The process of paper selection on clear with the inclusion and exclusion criteria.

This included the eligibility criteria and the methods used are valid and reliable.

There is indeed enough detail to replicate the study

Please change PubMed to PubMed/MEDLINE throughout the manuscript.

You did not use other keywords for specific arrhythmias except the general term arrhythmias? I think you might have missed some studies if so.

Results

● Is the data presented in an appropriate way?

● Tables and figures relevant and clearly presented?

● Appropriate units, rounding, and number of decimals?

● Titles, columns, and rows labelled correctly and clearly?

● Categories grouped appropriately?

● Does the text in the results add to the data or is it repetitive?

● Are you clear about what is a statistically significant result?

● Are you clear about what is a practically meaningful result?

The data is represented appropriately. However, the figures need to be labeled

appropriately with an appropriate title and interpretation text to make it easy for the

reader.

The text is not repetitive, but it needs to be clarified what is statistically significant

and what is a practically meaningful result.

If the association is not significant, then there is no association. There is no need to state that the association is non-significant.

Also, you did not state with which arrhythmias exposure to PM was associated with. arrhythmias is a very broad term...

Please specify both in the text and table.

Discussion and Conclusions

● Are the results discussed from multiple angles and placed into

context without being overinterpreted?

● Do the conclusions answer the aims of the study?

● Are the conclusions supported by references or results?

● Are the limitations of the study fatal or are they

opportunities to inform future research?

The results are discussed from multiple angles and used within context. The

conclusion does indeed answer the aims and is supported by references.

The limitations are well included in the study with possible recommendations

Overall

● Was the study design appropriate to answer the aim?

● What did this study add to what was already known on this topic?

● What were the major flaws of this article?

● Is the article consistent within itself?

The study design was appropriate, major flaws identified are included below.

Otherwise the article was consistent within itself

Overall strengths of the article and what impact it might have in your field

The paper will add impact to the scientific communication leaving opportunity for more studies around this topic.

Specific comments on weaknesses of the article and what could be done to improve it

- Review the grammar in the text

- Review the acceptable manner of labelling the figures in scientific writing

- please check comments listed above

6. PLOS authors have the option to publish the peer review history of their article (what does this mean?). If published, this will include your full peer review and any attached files.

Reviewer #1: No

Reviewer #2: No

---

## [Author Response · Author response to Decision Letter 0]

26 Feb 2024

Reviewers comment on the manuscript 

Title: Effect modification of diabetic status on the association between exposure to particulate matter and cardiac arrhythmias: a systematic review and meta-analysis [PONE-D-23-26753]

Dear Editor and reviewers,

 We would like to thank you for your valuable reviews and comments. It is our great pleasure to have an opportunity to revise our manuscript. We have revised and modified our manuscript with some additional information as suggested by reviewers’ comment. We hope that our revisions will improve the quality of the manuscript and give a clearer vision of research methodology to meet qualification for publication in PLOS ONE. Please inform us if further information or clarification is needed to be addressed.

Reviewer 1

Overall comments

• Overall, the methodological approach and execution of the systematic review and meta-analysis appear sound.

• Good use of Cochrane guidelines and pre-registration of the study.

• The manuscript would benefit from English language copyediting to correct some issues with grammar and flow.

• The small number of studies for meta-analysis (3 each for PM2.5 and PM10) calls into question the use of random effects model and whether it is sufficiently powered; recommend strengthening rationale for use of this model and explanation of the power expectations. (Although I2 is appropriately used to assess heterogeneity, please note the Cochrane caution, “Authors should recognize that there is much uncertainty in measures such as I2 and Tau2 when there are few studies. Thus, use of simple thresholds to diagnose heterogeneity should be avoided.”) [Chapter 10: Analysing data and undertaking meta-analyses | Cochrane Training]. Consider whether the number of studies is sufficient to justify the meta-analysis or if it should be dropped (for example, Chiu’s 1% weight in PM10 analysis is a vanishingly small contribution).

o Thank you for your comment on this very serious point. We realize that when only a small number of studies are included for meta-analysis, the random-effects model might not be appropriate, as the estimation of heterogeneity statistics to be used in weighting would be problematic. Our rationale was based on our prior assumption that all studies are heterogeneous in design, population included, and methodology. Therefore, we pre-specified our analysis model as a random-effects model without using any specific threshold for heterogeneity tests, like I-squared or Q test. In other words, it was all a context-based assumption. A recommendation to deal with the meta-analysis of a very small number of studies was suggested by Bender et al. in 2018 as follows: If heterogeneity is too large, no quantitative summary should be performed.

If the assumption seems to be valid that the same true effect is estimated in all studies (There is no true heterogeneity), a meta-analysis with common (fixed) effect should be applied.

When a certain amount of heterogeneity can be expected, the method of choice is a meta-analysis with random-effects. The recommended standard approach is the Knapp-Hartung method.

If the application of a frequentist random-effects model is warranted but not possible or most sensible (heterogeneity parameter not reliably estimable), conclusions on treatment effects should be based on the common (fixed) effect meta-analysis.

o In our case, we believe that there is true heterogeneity among studies and that we should not assume that there are the same true effects across studies. However, as it might not be sensible to apply a random-effects model due to unreliable estimation of heterogeneity parameters we therefore choose to perform an additional sensitivity analysis to examine the robustness of our conclusions by analysing all pooled results again using the HKSJ method and fixed effect model. If all the results and conclusions were consistent then it means that our results were robust. 

o From our analysis, it was found that the effects of particulate matter (regardless of their sizes) were significant in the overall population for all three analytic models. The association between PM of any size was also significantly associated with arrhythmia in both non-diabetic and diabetic subgroups when the DL random-effects model was used. When the HKSJ method was used, the results were similar in terms of the effect estimates but not statistically significant. The fixed-effect model provides statistically significant results for both the overall and the subgroup analysis. However, the pooled effects were quite different from the two random-effects models used. 

o For PM2.5, all three modeling approaches provide consistent results for the effect of PM2.5 and arrhythmia in the overall population, but not in the subgroup by diabetic status. For PM10, the associations identified from all three modeling techniques were not consistent in both the overall and subgroup by diabetic status.

o Revisions made to the manuscript:

o We added in the methods section as follows: We conducted a post hoc sensitivity analysis to examine the robustness of our pooled results in cases where the estimation of heterogeneity parameters might be unreliable due to the low number of studies included in the random-effects model. This was done by performing all analyses using both the Hartung-Knapp-Sidak-Jonkman (HKSJ) method for the random-effects model and the fixed-effect model as suggested in literature. Subsequently, we compared the pooled results obtained from the three methods. If all the results were consistent in terms of statistical significance, then our primary analyses using the DL random-effects model could be considered robust.

o We added in the results section as follows: It was found that the effects of PM of any size were significant in the overall population for all three analytic models (Table 2). The association between PM of any size was also significantly associated with arrhythmia in both non-diabetic and diabetic subgroups when the DL random-effects model was used (Fig). When the HKSJ method was used, the results were similar in terms of the effect estimates but not in terms of statistical significance (S1-S3 Fig). The fixed-effect model provides statistically significant results for both the overall and the subgroup analysis (S4 Fig). However, the pooled ORs were quite different from those of random-effects models used. For PM2.5, all three modeling approaches provide consistent results for the effect of PM2.5 and arrhythmia in the overall population (S5 Fig), but not in the subgroup by DM status (S6-S7 Fig). For PM10, the associations identified from all three modeling techniques were not consistent in both the overall and subgroup by diabetic status (S9-S12 Fig). 

We added in the limitations section as follows: Thirdly, performing a meta-analysis using a small number of studies threatens the validity of the statistical model, especially when the random-effects model was pre-specified. Alternative modeling methods were used to examine the robustness of the primary results and found that only a few aspects of the analyses were robust.

• The reporting of results is generally good, but when reporting the effects of both PM10 and PM2.5 on arrythmias, sometimes the language is unclear. Please specify how particulate exposure was combined when reporting these together. Furthermore, the manuscript would be strengthened by explaining why combining PM10 and PM2.5 results makes sense in the first place.

o Response to reviewer: Thank you for your comment on this point. For our analysis, the exposure status was defined as follows: exposure to PM of any size (combining both PM2.5 and PM10), exposure to PM2.5, and exposure to PM10. For the first type of exposure, the effects of both PM2.5 and PM10 would be identified. For the latter, the effects were specific to each pollutant. The rationale for the identification of the combined effects of PM2.5 and PM10 was that both PM2.5 and PM10 are particles that pollute the air. The only difference was in their diameter size. In the real environment, people are exposed to a mixture of particles of different sizes. Studying the combined effect reflects the complexity of actual exposure scenarios, making the research findings more relevant and better informing the development of public health policies and regulatory measures. Moreover, both have been found to be associated with cardiac arrhythmia, specifically atrial fibrillation (AF).

o Revisions made to the manuscript: 

o We added the following into the methods section of the manuscript for clarification: For our analysis, we defined exposure status as follows: exposure to PM of any size (combining both PM2.5 and PM10), exposure to PM2.5, and exposure to PM10. In the case of exposure to PM of any size, we would identify the effects of both PM2.5 and PM10. Considering that people in real environments are exposed to a mixture of the two, identifying PM in general, regardless of size, may be more relevant and better inform public health policy. 

We also revised the results section to clarify that we’re examining the combined effects of both PM2.5 and PM10 by using the terms “OM of any size”. 

• Publication bias is a notorious potential issue in meta-analysis; although the authors discuss searching for “grey literature,” it is not clear whether such studies were uncovered and how they were handled to eliminate bias. A funnel plot would be useful to elucidate.

o Response to reviewer: Thank you for your suggestions on this point. Firstly, no additional study was identified from grey literature search. We stated this in the results section of the manuscript. Secondly, we recognize that a funnel plot could be used to evaluate the extent of publication bias in meta-analysis. However, according to Cochrane’s suggestion, funnel plots and statistical tests for publication bias should not be used in a meta-analysis that includes fewer than 10 studies. In our study, only five studies were included in the analysis. However, these five studies were split into multiple sub-studies according to PM size. For example, one study reporting the odds ratio separately for PM2.5 and PM10 would be separated into two sub-studies. Only the pooled odds ratio of the association between PM (of any sizes) and arrhythmia in the overall population was derived from more than 10 studies. Thus, the forest plot would be done only for this outcome in the overall population. For other subgroup analyses that included fewer than 10 studies, we believe that it is more appropriate not to use the funnel plot or statistical test to evaluate the presence of publication bias. We understand that without evaluation, we cannot conclude that there is no publication bias. Therefore, we acknowledge this point in the limitation section of the manuscript.

o Revisions made to the manuscript: 

We added the following sentence to the results section: No additional study was identified from grey literature search. 

We added in the methods section our planned analysis: “Publication bias would be assessed with a funnel plot and Egger’s test if the number of included studies were higher or equal to ten. In the case where publication bias is likely, the Trim-and-fill method would be used to assess and adjust the pooled results for publication bias.”

We added in the limitations section: Secondly, our pooled results were derived from a relatively small number of studies, which affects the precision of the estimates. Additionally, we could not test or confirm the absence of publication bias in our results.

• The manuscript would benefit from a discussion of type 1 & 2 DM, whether they play differing roles in PM effects, and how this is reflected (or deemed irrelevant) in these findings.

o Response to reviewer: Thank you very much for your comments. We agree with the reviewer that this is actually a very interesting point to discuss. There were several reports on the risk of cardiac arrhythmias, specifically atrial fibrillation, in patients with both type 1 and type 2 diabetes. Moreover, there were also studies that proposed the potential of exposure to particulate matter as an etiologic factor for both type 1 and type 2 diabetes. However, whether the effect of PM on the occurrence of arrhythmias differs between these two types of diabetes is yet unknown. We would like to clarify that we did not treat these two types of DM as equivalent. When we conceptualized our review protocol, we aimed to set up a broader question to gather wider evidence, including both type 1 and type 2 diabetes. Thus, we did not specify an inclusion to be restricted to only a single type of diabetes. If, after we had gathered all evidence for the review, there was a sufficient amount of data to perform subgroup analysis by DM types, then this would be interesting. Unfortunately, this was not the case in our review. We found that for all the full-texts we gathered for the review, they all provided non-specific definitions for DM and did not report separate analyses by DM types.

o Revisions made to the manuscript: 

o We revised the methods section as follows: population with and without DM (regardless of DM types) and report the outcome data separately for each group

o We revised Table 1 by adding the definition of DM of each included paper in the review and provided descriptions as follows: Most studies did not provide detailed data on DM diagnostic criteria or information on DM types. Three studies defined DM based on ICD-9 and ICD-10 codes.

o We acknowledged this point as one of our limitations as follows: Fourthly, due to the scarcity of the data reported in original articles, we could not perform the analysis to examine the differential effects for each type of DM.

• Methodologically, the use of imputation and final missingness are an important detail that receives no detailed description.

o Revisions made to the manuscript: We revised the methods section to be clearer as follows: For studies with incomplete data required for the analysis, we contacted the authors of those studies via email. If they did not respond within 2 weeks, the data would be imputed or reported as missing. WebPlotDigitizer was used to extract numeric data from graphs where appropriate. 

o We also added the findings in the results section as follows: All the data required for analysis were reported or visualized, and no imputation of data was made during the extraction process. The association parameters for the following studies were digitized from graphs: Lee (2019), Zanobetti (2014), Zheng (2018), Liang (2021), and Jin (2022).

Specific sections

Abstract and Introduction

• Line 30: please specify the timeframe in which DM diagnoses have increased

o Response to reviewer: Thank you for your comment. However, during revision, the introduction paragraphs were revised according to suggestions by another reviewer and this sentence was removed.

o Revisions made to the manuscript: We have revised most part of the introduction of the manuscript. The sentence was removed.

• Line 54: please provide citations for the studies mentioned in “Additionally, several studies were performed to investigate the effect of PM on cardiac arrhythmias, and some of these continued studying to identify the diseases or conditions that were sensitive to the effect of PM such as hypertension and diabetes mellitus (DM).

o Response to reviewer: Thank you for your comment. However, during revision, the introduction paragraphs were revised according to suggestions by another reviewer and this sentence was removed.

o Revisions made to the manuscript: We have revised most part of the introduction of the manuscript. The sentence was removed.

• Line 58 Seems like a repeat.

o Response to reviewer: Thank you for your comment. However, during revision, the introduction paragraphs were revised according to suggestions by another reviewer and this sentence was removed.

o Revisions made to the manuscript: We have revised most part of the introduction of the manuscript. The sentence was removed.

• Line 63: It seems that your hypothesis is that DM diagnosis indicates a more susceptible group.

o Response to reviewer: Thank you for your comment. However, during revision, the introduction paragraphs were revised according to suggestions by another reviewer and this sentence was modified.

o Revisions made to the manuscript: adding more. It now reads “As the pathogenic mechanism of arrhythmia from both PM exposure and DM seemed to overlap, and there was previous evidence showing that exposure to PM increased the risk of arrhythmia in both DM-induced rats and diabetic patients [16,17], it was hypothesized that the effect of PM on cardiac arrhythmia might be more pronounced in individuals with DM than in individuals without DM [10,12,16,17].”

• Line 65: Are you suggesting that AF is not a type of arrhythmia? It is unclear why you highlight this fact.

o Response to reviewer: Thank you for your suggestion. We were not suggesting that AF is not one of cardiac arrhythmia type. We modified the sentence to make our idea clearer. 

o Revisions made to the manuscript: It now reads “There were meta-analyses that investigated the association between PM and cardiac arrhythmias, but almost all of them focused specifically on AF, which is only one type of cardiac arrhythmia. Furthermore, none of them performed a subgroup analysis of diabetic status [18–22].”

Method

• Line 76: Given different disease trajectory and effects, discuss type 1 versus type 2 diabetes and why you treat them as equivalent.

o Response to reviewer: Thank you very much for your comments. We agree with the reviewer that this is actually a very interesting point to discuss. There were several reports on the risk of cardiac arrhythmias, specifically atrial fibrillation, in patients with both type 1 and type 2 diabetes. Moreover, there were also studies that proposed the potential of exposure to particulate matter as an etiologic factor for both type 1 and type 2 diabetes. However, whether the effect of PM on the occurrence of arrhythmias differs between these two types of diabetes is yet unknown. We would like to clarify that we did not treat these two types of DM as equivalent. When we conceptualized our review protocol, we aimed to set up a broader question to gather wider evidence, including both type 1 and type 2 diabetes. Thus, we did not specify an inclusion to be restricted to only a single type of diabetes. If, after we had gathered all evidence for the review, there was a sufficient amount of data to perform subgroup analysis by DM types, then this would be interesting. Unfortunately, this was not the case in our review. We found that for all the full-texts we gathered for the review, they all provided non-specific definitions for DM and did not report separate analyses by DM types.

o Revisions made to the manuscript: 

o We revised the methods section as follows: population with and without DM (regardless of DM types) and report the outcome data separately for each group

o We revised Table 1 by adding the definition of DM of each included paper in the review and provided descriptions as follows: Most studies did not provide detailed data on DM diagnostic criteria or information on DM types. Three studies defined DM based on ICD-9 and ICD-10 codes.

o We acknowledged this point as one of our limitations as follows: Fourthly, due to the scarcity of the data reported in original articles, we could not perform the analysis to examine the differential effects for each type of DM.

• Line 85: What became of your grey literature search? Were studies uncovered and included in this analysis? How does your later discussion of bias relate to this search?

o Response to reviewer: Thank you for your concern on this important point. However, no additional study was identified from grey literature search. We stated this in the results section of the manuscript. 

o Revisions made to the manuscript: Add this sentence to the results section: No additional study was identified from grey literature search.

• Line 101: Here and elsewhere the “types of effect” should be “type (or length) of exposure”. For reference, see your bibliography for Kim et al (2019) where “long-term” is clearly referring to PM exposure: Kim IS, Yang PS, Lee J, Yu HT, Kim TH, Uhm JS, et al. Long-term exposure of fine particulate matter air pollution and incident atrial fibrillation in the general population: A nationwide cohort study.

o Response to reviewer: Thank you for your suggestion. We have revised accordingly.

o Revisions made to the manuscript: changed to “type (or length) of exposure (i.e., short, or long term)”

• Line 105: Provide substantially more detail on the methods for imputation, which data were imputed, and what the remaining missingness was. This is an extremely important detail lacking clarity.

o Response to reviewer: Thank you for your comments on this point. Only the data required for the analysis that were not reported would be requested from the authors. However, if the data could be calculated indirectly from other surrounding figures or extracted from graphs, it would be done. We used a web application called WebPlotDigitizer to extract numeric data from graphs where appropriate. In this review, all the data required for analysis were reported or visualized, and no imputation of data was made during the extraction process. The association parameters for the following studies were digitized from graphs: Lee (2019), Zanobetti (2014), Zheng (2018), Liang (2021), and Jin (2022).

o Revisions made to the manuscript: We revised the methods section to be clearer as follows: For studies with incomplete data required for the analysis, we contacted the authors of those studies via email. If they did not respond within 2 weeks, the data would be imputed or reported as missing. WebPlotDigitizer was used to extract numeric data from graphs where appropriate. 

o We also added the findings in the results section as follows: All the data required for analysis were reported or visualized, and no imputation of data was made during the extraction process. The association parameters for the following studies were digitized from graphs: Lee (2019), Zanobetti (2014), Zheng (2018), Liang (2021), and Jin (2022).

• Line 106 and following – risk of bias: Consider the use of a funnel plot to further elucidate bias handling.

o Response to reviewer: Thank you for your suggestions on this point. We recognize that a funnel plot could be used to evaluate the extent of publication bias in meta-analysis. However, according to Cochrane’s suggestion, funnel plots and statistical tests for publication bias should not be used in a meta-analysis that includes fewer than 10 studies. In our study, only five studies were included in the analysis. However, these five studies were split into multiple sub-studies according to PM size. For example, one study reporting the odds ratio separately for PM2.5 and PM10 would be separated into two sub-studies. Only the pooled odds ratio of the association between PM (of any sizes) and arrhythmia in the overall population was derived from more than 10 studies. Thus, the forest plot would be done only for this outcome in the overall population. For other subgroup analyses that included fewer than 10 studies, we believe that it is more appropriate not to use the funnel plot or statistical test to evaluate the presence of publication bias. We understand that without evaluation, we cannot conclude that there is no publication bias. Therefore, we acknowledge this point in the limitation section of the manuscript.

o Revisions made to the manuscript: 

We added in the methods section our planned analysis: “Publication bias would be assessed with a funnel plot and Egger’s test if the number of included studies were higher or equal to ten. In the case where publication bias is likely, the Trim-and-fill method would be used to assess and adjust the pooled results for publication bias.”

We added in the limitations section: Secondly, our pooled results were derived from a relatively small number of studies, which affects the precision of the estimates. Additionally, we could not test or confirm the absence of publication bias in our results.

• Lines 122,123: Please provide more details on your rationale for using a random effects model. Given the small number of studies used for meta-analysis, it is not clear how you evaluated I2 and Q or compared results to a fixed-effects methodology, and whether the small sample size impacts potential odds ratio estimates.

o Response to reviewer: Thank you for your comment on this very serious point. We realize that when only a small number of studies are included for meta-analysis, the random-effects model might not be appropriate, as the estimation of heterogeneity statistics to be used in weighting would be problematic. Our rationale was based on our prior assumption that all studies are heterogeneous in design, population included, and methodology. Therefore, we pre-specified our analysis model as a random-effects model without using any specific threshold for heterogeneity tests, like I-squared or Q test. In other words, it was all a context-based assumption. A recommendation to deal with the meta-analysis of a very small number of studies was suggested by Bender et al. in 2018 as follows: If heterogeneity is too large, no quantitative summary should be performed.

If the assumption seems to be valid that the same true effect is estimated in all studies (There is no true heterogeneity), a meta-analysis with common (fixed) effect should be applied.

When a certain amount of heterogeneity can be expected, the method of choice is a meta-analysis with random-effects. The recommended standard approach is the Knapp-Hartung method.

If the application of a frequentist random-effects model is warranted but not possible or most sensible (heterogeneity parameter not reliably estimable), conclusions on treatment effects should be based on the common (fixed) effect meta-analysis.

o In our case, we believe that there is true heterogeneity among studies and that we should not assume that there are the same true effects across studies. However, as it might not be sensible to apply a random-effects model due to unreliable estimation of heterogeneity parameters we therefore choose to perform an additional sensitivity analysis to examine the robustness of our conclusions by analysing all pooled results again using the HKSJ method and fixed effect model. If all the results and conclusions were consistent then it means that our results were robust. 

o From our analysis, it was found that the effects of particulate matter (regardless of their sizes) were significant in the overall population for all three analytic models. The association between PM of any size was also significantly associated with arrhythmia in both non-diabetic and diabetic subgroups when the DL random-effects model was used. When the HKSJ method was used, the results were similar in terms of the effect estimates but not statistically significant. The fixed-effect model provides statistically significant results for both the overall and the subgroup analysis. However, the pooled effects were quite different from the two random-effects models used. 

o For PM2.5, all three modeling approaches provide consistent results for the effect of PM2.5 and arrhythmia in the overall population, but not in the subgroup by diabetic status. For PM10, the associations identified from all three modeling techniques were not consistent in both the overall and subgroup by diabetic status.

o Revisions made to the manuscript:

o We added in the methods section as follows: We conducted a post hoc sensitivity analysis to examine the robustness of our pooled results in cases where the estimation of heterogeneity parameters might be unreliable due to the low number of studies included in the random-effects model. This was done by performing all analyses using both the Hartung-Knapp-Sidak-Jonkman (HKSJ) method for the random-effects model and the fixed-effect model as suggested in literature. Subsequently, we compared the pooled results obtained from the three methods. If all the results were consistent in terms of statistical significance, then our primary analyses using the DL random-effects model could be considered robust.

o We added in the results section as follows: It was found that the effects of PM of any size were significant in the overall population for all three analytic models (Table 2). The association between PM of any size was also significantly associated with arrhythmia in both non-diabetic and diabetic subgroups when the DL random-effects model was used (Fig). When the HKSJ method was used, the results were similar in terms of the effect estimates but not in terms of statistical significance (S1-S3 Fig). The fixed-effect model provides statistically significant results for both the overall and the subgroup analysis (S4 Fig). However, the pooled ORs were quite different from those of random-effects models used. For PM2.5, all three modeling approaches provide consistent results for the effect of PM2.5 and arrhythmia in the overall population (S5 Fig), but not in the subgroup by DM status (S6-S7 Fig). For PM10, the associations identified from all three modeling techniques were not consistent in both the overall and subgroup by diabetic status (S9-S12 Fig). 

We added in the limitations section as follows: Thirdly, performing a meta-analysis using a small number of studies threatens the validity of the statistical model, especially when the random-effects model was pre-specified. Alternative modeling methods were used to examine the robustness of the primary results and found that only a few aspects of the analyses were robust.

Results

• Line 149: Please explain why mixing long-term and short-term exposure durations (what you call “effect”) is appropriate for the meta-analysis.

o Response to reviewer: Thank you for your comment on this point. Two studies examined the long-term effects of PM, whereas the rest examined the short-term effects as shown in Table 1. For the five studies that we included for meta-analysis, all addressed the short-term effects. Therefore, we believe that this would not affect the analysis. However, to make our conclusion more accurate, we would state that the identified effects were short-term, not long-term.

o Revisions made to the manuscript:

We added in the results section the following sentences: All five studies focused on short-term effects of PM exposure. 

We also modified the way we wrote the results and conclusions to clarify that the effects identified from our meta-analysis is the short-term effect.

• Line 169 and Table 1: You mention “9 studies included in the meta-analysis”, but in Figure 1 I see only 6 studies that contribute to findings. (For example, where is Jin et al. in Figure 1?)

o Response to reviewer: Thank you for your comment on this point. We stated that “According to 5 studies which were included in our meta-analysis.” So, there were only five studies that provided a sufficient amount of data to be included in the analysis, and these studies were conducted by Peel (2006), Chiu (2009), Zanobetti (2014), Lee (2019), and Dahlquist (2020). The remaining four papers, including the one by Jin et al., were not included in the analysis.

o Revisions made to the manuscript: We revised the results section as follows: According to the five studies included in our meta-analysis [28,29,32,34,35], a total of 4,431,452 participants were included, with 2,556 of them being diabetic patients. The average age of participants was 74 years old. The studies employed various outcome measurements, with two focusing on hospital admissions for cardiac arrhythmias [28,32,36], two utilizing ECG measurements [29,35], and one centering on emergency room visits for cardiac arrhythmias [34]. All five studies focused on short-term effects of PM exposure.

• Line 175: please clarify that this section is addressing the effect of both (or either) PM2.5 and PM10 together, whereas the subsequent sections are specific to each pollutant alone. 

o Response to reviewer: Thank you for your suggestions. We revised accordingly. 

o Revisions made to the manuscript: 

o We added the following into the methods section of the manuscript for clarification: For our analysis, we defined exposure status as follows: exposure to PM of any size (combining both PM2.5 and PM10), exposure to PM2.5, and exposure to PM10. In the case of exposure to PM of any size, we would identify the effects of both PM2.5 and PM10. Considering that people in real environments are exposed to a mixture of the two, identifying PM in general, regardless of size, may be more relevant and better inform public health policy.

o We also revised the results section to clarify that we’re examining the combined effects of both PM2.5 and PM10. 

Discussion

• Lines 234,235: Please restate your findings with respect to the effect modification, as you did in the preceding paragraph.

o Response to reviewer: Thank you for your comment on this point. We have revised most part of the discussion section according to the suggestions made by reviewers. Therefore, most of the sentences were modified. However, we made sure that all your comments were addressed within the new version of the manuscript.

• Line 240: Please expand on the comparability thesis as the reason for differing Schneider results.

o Response to reviewer: Thank you for your comment on this point. We have revised most part of the discussion section according to the suggestions made by reviewers. Therefore, most of the sentences were modified. However, we made sure that all your comments were addressed within the new version of the manuscript.

• Lines 258-260: The selection criteria are an extremely important point and should probably be included in your abstract, results, and other text that states a finding: “…among PM and cardiac arrythmia studies that include diabetes as a potential modifier, our findings were…”

o Response to reviewer: Thank you for your comment on this point. We have revised most part of the discussion section according to the suggestions made by reviewers. Therefore, most of the sentences were modified. However, we made sure that all your comments were addressed within the new version of the manuscript.

 

Reviewer 2

Abstract, title and references

o The aim is clear; however, it may be ideal to add to the population the study is targeting. The same applies to the title. It is clearly stated what was found in the study. The references are relevant, recent, and used correctly.

Response to reviewer: Thank you for your comment. Our review was intended to answer the question of effect modification of diabetic status on the association between exposure to PM and cardiac arrhythmia in a general population. Therefore, we revised the title and the abstract accordingly.

Revisions made to the manuscript: The title was revised to: Effect modification of diabetic status on the association between exposure to particulate matter and cardiac arrhythmias in a general population: a systematic review and meta-analysis

Introduction/background

o It is clear what is already known about the topic and the research question is clearly outlined and justified given the limited information on what is already known about the topic.

o However, I find the introduction too short. The authors discuss oxidative stress and inflammation but this has little to do with the article itself. The authors should discuss more about PM, its association with arrhythmias and which arrhythmias in particular, the impact of diabetes on arrhythmias, the impact of PM on diabetes etc.

Response to reviewer: Thank you for your comment. We have re-written the introduction paragraphs according to your suggestions.

Methods

o The process of paper selection on clear with the inclusion and exclusion criteria. This included the eligibility criteria and the methods used are valid and reliable. There is indeed enough detail to replicate the study

o Please change PubMed to PubMed/MEDLINE throughout the manuscript.

Response to reviewer: Thank you for your comment. We have revised accordingly.

Revisions made to the manuscript: Changed PubMed to PubMed/MEDLINE throughout the manuscript.

o You did not use other keywords for specific arrhythmias except the general term arrhythmias? I think you might have missed some studies if so.

Response to reviewer: Thank you for your comment on this point. Since our review question focuses on any type of cardiac arrhythmia, not just one specific type, we used the term ‘cardiac arrhythmias’ as one of our keywords. We believe that this term leads to a more sensitive search strategy and complies more with our review question.

Revisions made to the manuscript: No changes made.

Results

o The data is represented appropriately. However, the figures need to be labeled appropriately with an appropriate title and interpretation text to make it easy for the reader.

Response to reviewer: Thank you for your comments. We revised the figures as suggested.

o The text is not repetitive, but it needs to be clarified what is statistically significant and what is a practically meaningful result. If the association is not significant, then there is no association. There is no need to state that the association is non-significant.

Response to reviewer: Thank you for your comments. We revised the figures as suggested.

Revisions made to the manuscript: We revised accordingly.

o Also, you did not state with which arrhythmias exposure to PM was associated with. arrhythmias are a very broad term...

Response to reviewer: Thank you for your comment on this point. We agree with the reviewer that this is a crucial consideration. However, since our review question focuses on the broader association between PM exposure and any type of cardiac arrhythmia, and we have included studies addressing various types of cardiac arrhythmia in our review and analysis, we can only interpret the association as 'arrhythmias' in general and cannot be more specific. Among the five studies included in the meta-analysis, two specified the outcome according to ICD9 code 427, which is dysrhythmia in general; one study defined arrhythmia episodes measured as ventricular ectopy on EKG, and another two focused only on atrial fibrillation (AF). Therefore, there is a mix in the outcome definitions used across studies.

Revisions made to the manuscript: 

We have stated in the last introduction paragraph that our review question focused on any cardiac arrhythmias as follows: We then aimed to evaluate the effect of PM on any cardiac arrhythmias and to compare the occurrence of any cardiac arrhythmias in the presence of PM between diabetic and non-diabetic patients.

We acknowledged this point as one of our limitations as follows: In terms of the overall characteristics of the included studies, significant demographic and methodological heterogeneity was observed. The definitions of DM and arrhythmia varied, and the interpretation of results should be made carefully.

Discussion and Conclusions

o The results are discussed from multiple angles and used within context. The conclusion does indeed answer the aims and is supported by references.

o The limitations are well included in the study with possible recommendations

Overall

o Otherwise, the article was consistent within itself

o Overall strengths of the article and what impact it might have in your field

o The paper will add impact to scientific communication leaving opportunity for more studies around this topic.

---

## [Decision Letter · Decision Letter 1]

24 Mar 2024

Effect modification of diabetic status on the association between exposure to particulate matter and cardiac arrhythmias in a general population: a systematic review and meta-analysis

PONE-D-23-26753R1

Dear Dr. Phinyo,

We’re pleased to inform you that your manuscript has been judged scientifically suitable for publication and will be formally accepted for publication once it meets all outstanding technical requirements.

Kind regards,

Trenton Honda

Academic Editor

PLOS ONE

Reviewers' comments:

Reviewer's Responses to Questions

**Comments to the Author**

1. If the authors have adequately addressed your comments raised in a previous round of review and you feel that this manuscript is now acceptable for publication, you may indicate that here to bypass the “Comments to the Author” section, enter your conflict of interest statement in the “Confidential to Editor” section, and submit your "Accept" recommendation.

Reviewer #1: All comments have been addressed

Reviewer #2: All comments have been addressed

2. Is the manuscript technically sound, and do the data support the conclusions?

Reviewer #1: Yes

Reviewer #2: Yes

3. Has the statistical analysis been performed appropriately and rigorously? 

Reviewer #1: Yes

Reviewer #2: I Don't Know

4. Have the authors made all data underlying the findings in their manuscript fully available?

Reviewer #1: No

Reviewer #2: Yes

5. Is the manuscript presented in an intelligible fashion and written in standard English?

Reviewer #1: Yes

Reviewer #2: Yes

6. Review Comments to the Author

Reviewer #1: This is a good revision that addresses the questions I posed in the draft manuscript. I think this work will be a helpful addition to the literature.

Reviewer #2: The authors have satisfactorily answered my queries and the paper seems worthy of publication in the current form

7. PLOS authors have the option to publish the peer review history of their article (what does this mean?). If published, this will include your full peer review and any attached files.

Reviewer #1: No

Reviewer #2: No

---

## [Editor Report · Acceptance letter]

1 Apr 2024

PONE-D-23-26753R1 

PLOS ONE

Dear Dr. Phinyo, 

I'm pleased to inform you that your manuscript has been deemed suitable for publication in PLOS ONE. Congratulations! Your manuscript is now being handed over to our production team.

Kind regards, 

on behalf of

Dr. Trenton Honda 

Academic Editor

PLOS ONE